# Impact of nonrandom selection mechanisms on the causal effect estimation for two-sample Mendelian randomization methods

Yuanyuan Yu[1,2☯], Lei Hou[1,2☯], Xu Shi[3], Xiaoru Sun[1,2], Xinhui Liu[1,2], Yifan Yu[1,2], Zhongshang Yuan[1,2], Hongkai Li[1,2¤*], Fuzhong Xue[1,2¤*]

**1** Department of Biostatistics, School of Public Health, Cheeloo College of Medicine, Shandong University, Jinan, People's Republic of China, **2** Institute for Medical Dataology, Cheeloo College of Medicine, Shandong University, Jinan, People's Republic of China, **3** Department of Biostatistics, University of Michigan, Ann Arbor, Michigan, United States of America

☯ These authors contributed equally to this work.
¤ Current address: Institute for Medical Dataology, Cheeloo College of Medicine, Shandong University, Jinan, People's Republic of China
* lihongkaiyouxiang@163.com (HL); xuefzh@sdu.edu.cn (FX)

**Data Availability Statement:** The authors confirm that all data underlying the findings are fully available without restriction. Codes to implement the method and reproduce all simulations and

## Abstract

Nonrandom selection in one-sample Mendelian Randomization (MR) results in biased estimates and inflated type I error rates only when the selection effects are sufficiently large. In two-sample MR, the different selection mechanisms in two samples may more seriously affect the causal effect estimation. Firstly, we propose sufficient conditions for causal effect invariance under different selection mechanisms using two-sample MR methods. In the simulation study, we consider 49 possible selection mechanisms in two-sample MR, which depend on genetic variants ($G$), exposures ($X$), outcomes ($Y$) and their combination. We further compare eight pleiotropy-robust methods under different selection mechanisms. Results of simulation reveal that nonrandom selection in sample II has a larger influence on biases and type I error rates than those in sample I. Furthermore, selections depending on $X$ $+Y$, $G+Y$, or $G+X+Y$ in sample II lead to larger biases than other selection mechanisms. Notably, when selection depends on $Y$, bias of causal estimation for non-zero causal effect is larger than that for null causal effect. Especially, the mode based estimate has the largest standard errors among the eight methods. In the absence of pleiotropy, selections depending on $Y$ or $G$ in sample II show nearly unbiased causal effect estimations when the casual effect is null. In the scenarios of balanced pleiotropy, all eight MR methods, especially MR-Egger, demonstrate large biases because the nonrandom selections result in the violation of the Instrument Strength Independent of Direct Effect (InSIDE) assumption. When directional pleiotropy exists, nonrandom selections have a severe impact on the eight MR methods. Application demonstrates that the nonrandom selection in sample II (coronary heart disease patients) can magnify the causal effect estimation of obesity on HbA1c levels. In conclusion, nonrandom selection in two-sample MR exacerbates the bias of causal effect estimation for pleiotropy-robust MR methods.

analyses are available in the S9 Text. In our application, GWAS summary data for BMI, WHR and WHRadjBMI are from Genetic Investigation of ANthropometric Traits (GIANT), which can be download from GIANT consortium (https://portals.broadinstitute.org/collaboration/giant/index.php/GIANT_consortium_data_files) or MR base platform (https://www.mrbase.org/). Genetic data and individual data for HbA1c can be obtained in UK Biobank (https://www.ukbiobank.ac.uk/). We calculated the GWAS summary data for HbA1c in CHD patients and general population. All the GWAS summary data used in this paper can be found in S1 Data.

**Funding:** F.X. was supported by the National Natural Science Foundation of China (Grant 81773547) and Shandong Provincial Natural Science Foundation of China (ZR2019ZD02). H.L. was supported by the National Natural Science Foundation of China (Grant 82003557). Z.Y. was supported by Shandong Provincial Key Research and Development project (2018CXGC1210). The funders had no role in study design, data collection and analysis, decision to publish, or preparation of the manuscript.

**Competing interests:** The authors have declared that no competing interests exist.

## Author summary

It is well known that nonrandom selection in one-sample Mendelian Randomization (MR) can result in biased estimates and inflated type I error rates. Actually, two-sample MR analyses are more prone to be affected by nonrandom selection than one-sample MR analyses, because two samples for genome-wide association studies (GWAS) may be selected each under different mechanisms from the source population. Summary-level genetic association statistics in two-sample MR may be derived from different study designs such as case-control, case-only and cohort studies, which further inevitably affect the causal effect estimation of exposure on outcome. In this study, we firstly propose a theorem for causal effect invariance under different selection mechanisms. In the simulation, we design 49 combinations of nonrandom selection mechanisms in sample I and sample II, which are widespread in practical applications. The simulation results reveal that the selection mechanisms in sample II have a larger influence on biases and type I error rates than those in sample I. As an illustrative example, we find the nonrandom selection in sample II (coronary heart disease patients) can magnify the causal effect estimation of obesity on the HbA1c levels.

## 1 Introduction

Mendelian randomization (MR) uses genetic variants as instrumental variables (IV) to obtain an unbiased causal effect estimation in the presence of unmeasured confounding [1]. MR analysis assumes that the genetic variants satisfy instrumental variable assumptions including IV Relevance (the IV must be robustly associated with the exposure), IV Independence (the IV must be independent of unmeasured confounders), and Exclusion restriction (the IV must not have a direct effect on the outcome that is not mediated by the exposure) [2–4]. Two-sample MR leverages the summary-level genetic associations of exposure and outcome from two non-overlapping datasets to estimate causal effect. These summary-level genetic associations can be obtained from published literature provided by consortia of genome-wide association studies (GWAS), or directly from individual-level participant data [5, 6].

Nonrandom selection in one-sample MR [7] can result in biased estimates and inflated type I error rates. The magnitude of bias can differ according to the strength of instruments, the complexity of exposure-instrument association, and the nature of exposure effects. When selection depends on instrumental variables ($G$), conducting the analysis on the selected sample does not lead to biased estimates because, as shown in Fig 1, $G$ is not a collider (or a descendant of a collider) and thus does not induce a new open path between $G$ and exposure ($X$) or $G$ and outcome ($Y$). In contrast, because $X$ is a collider in the path $G \rightarrow X \leftarrow U \rightarrow Y$, when selection depends on $X$, conditioning on selection opens up a new path between $G$ and $U$, which violates the IV Independence assumption. Similarly, when there is a causal effect of $X$ on $Y$, $Y$ is also a descendant of collider $X$, and selection that depends on $Y$ will also open this path. Thus selection depending on $Y$ will induce bias only for the non-null causal effect of $X$ on $Y$ [8]. When there is a null causal effect of $X$ on $Y$, $Y$ is no longer a descendant of collider $X$, and nonrandom selection based on $Y$ does not induce any bias; hence there is no type I error inflation. Therefore, when selection depends on $X$, $Y$, or $X$ and $Y$, estimates of the causal effect are biased [7].

For two-sample MR analyses, we assume that genetic association statistics with the exposure (e.g. beta-coefficient $\hat{\beta}_{X_j}$ and standard error (SE) $\hat{\sigma}_{X_j}$) are obtained from sample I and the genetic association statistics with the outcome (e.g. beta-coefficient $\hat{\beta}_{Y_j}$ and SE $\hat{\sigma}_{Y_j}$) are obtained from

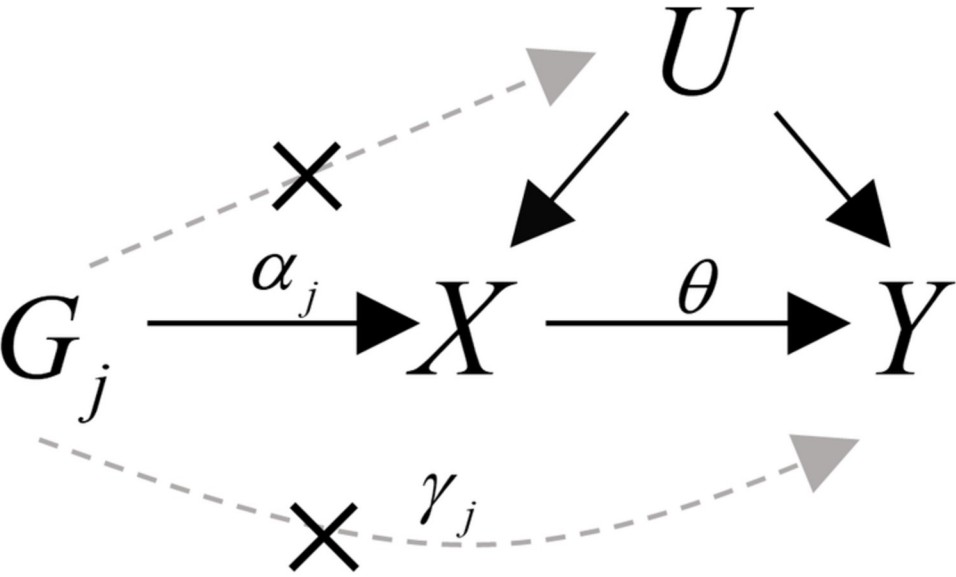

**Fig 1. Illustrative diagram of Mendelian randomization.** $G_j$, $j$-th genetic variant, with effect $\phi_j$ on confounders $U$, a direct effect $\alpha_j$ on the exposure $X$ and a direct effect $\gamma_j$ on the outcome $Y$.

sample II. Whether sample I or sample II is a random sample of the target population requires more attention. Two-sample MR analyses are more prone to be affected by nonrandom selection than one-sample MR analyses, in that two heterogeneous samples can be selected each under a different mechanism that differs from the source population [7]. Such sampling mechanisms include willingness to participate and survival to the participation date [1, 7]. This could lead to the violation of MR assumption. Recent IV-based estimators in MR studies include consensus methods, regression-based methods, likelihood-based methods, outlier-robust methods [9–16], which focus on relaxing the Exclusion restriction. Pleiotropy is characterized by genetic variants associated with multiple phenotypic variables that is common in MR studies and can lead to the violation of the Exclusion restriction. The extent to which these pleiotropy-robust methods can affect causal effect estimations under different selection mechanisms remains unclear.

In this study, we first propose sufficient conditions for causal effect invariance under different selection mechanisms using two-sample MR methods. We then consider 49 possible selection mechanisms in two-sample MR, which depend on $G$, $X$, $Y$ and their combination, respectively. In the simulation, we compare eight pleiotropy-robust methods under different selection mechanisms. Finally, we use an application to explore the extent to which nonrandom selection influences the estimation of the causal effect of obesity on HbA1c levels.

## 2 Materials and methods

### 2.1 Modeling assumptions and summary level data

Let $G = \{G_1, G_2, \ldots, G_J\}$ denote $J$ genetic variants that are mutually independent, and $X$, $Y$ and $U$ be the exposure, outcome and unmeasured confounder, respectively. We assume the model of Fig 1 is:

$$E[Y|X, U, G_1, \ldots, G_J] = \theta X + U + \sum_{j=1}^{J} \gamma_j G_j$$

$$E[X|U, G_1, \ldots, G_J] = \sum_{j=1}^{J} \alpha_j G_j + U. \tag{1}$$

Each of the valid genetic variants must satisfy the three IV assumptions (Relevance, Independence and Exclusion restriction), as well as linearity, homogeneity, monotonicity and non-overlap assumptions [11, 17].

**Linear and homogeneity assumption.**   Estimating the causal effect of $X$ on $Y$ in the full study population requires linearity of the IV-$X$, IV-$Y$ and $X$-$Y$ relationships. There is no effect heterogeneity in the $X$-$Y$ relationships. Linearity for the IV-$X$ association is necessary for point estimates but not for testing the null hypothesis.

**Monotonicity assumption.**   Monotonicity in the context of MR means that increasing the number of effect alleles for an individual can only increase the exposure.

**No sample overlap.**   Two-sample MR requires two non-overlapping samples to estimate causal effects. MR analyses using IV-$X$ and IV-$Y$ associations in the same sample or in partially overlapping samples may be prone to weak instrument bias towards the $X$-$Y$ estimate that would be obtained using conventional methods. A simulation suggested that bias due to sample overlap is a linear function of the proportion of overlap between samples [17]. More details of modeling assumptions are shown in S1 Text.

## 2.2 Sufficient conditions for causal effect invariance

Our objective is to calculate the causal effect $\beta_{G_jY}/\beta_{G_jX}$ of $X$ on $Y$ using two-sample MR methods at the population-level distribution (i.e., Wald ratio method). The genetic association statistics with exposure ($\hat{\beta}_{G_jX}$, $j = 1,2,\ldots,J$) and outcome ($\hat{\beta}_{G_jY}$, $j = 1,2,\ldots,J$) can be obtained from sample I and sample II, respectively. Sample I and sample II are random samples from population I and population II, respectively. $S_1$ and $S_2$ are binary variables indicating whether a participant is selected or unselected in sample I and sample II, respectively. Restricting the analysis to the selected sample implies conditioning on $S_1$ or $S_2$ equal to one, which is represented by a box around $S_1$ or $S_2$. Due to the preferential selection, the estimation is $\hat{\beta}_{G_jY|S_2=1}/\hat{\beta}_{G_jX|S_1=1}$. The natural question to ask is under what conditions the causal effect can be recovered by sample I and II with preferential selection, that is, $\beta_{G_jY}/\beta_{G_jX} = \beta_{G_jY|S_2=1}/\beta_{G_jX|S_1=1}$, and the extent to which selection may affect the causal effect estimate.

Based on the classical instrumental variable assumptions [2–4], we propose the following theorem to explore the sufficient conditions for causal effect invariance in two-sample MR based on the Wald ratio method.

**Theorem 1** The sufficient conditions for causal effect invariance under different selection mechanisms $\beta_{G_jY|S_2}/\beta_{G_jX|S_1} = \beta_{G_jY}/\beta_{G_jX}$ from two populations are:

a.  for each valid instrumental variable $G_j$, $S_1 \perp G_j$ or $S_1 \perp X|G_j$ in population I and $S_2 \perp G_j$ or $S_2 \perp Y|G_j$ in population II, respectively, or

b.  $G_j \perp Y|S_2$ and $G_j \perp Y$ for each valid instrumental variable in population II.

Theorem 1 provides sufficient conditions for causal effect invariance under different selection mechanisms using two-sample MR methods. We also provide Directed Acyclic Graphs (DAGs) that satisfied condition (a-b) in **Theorem 1** (Figs 2 and 3). Three scenarios including no nonrandom selection mechanism, selection depending on unmeasured confounders and selection depending on genetic variants (Fig 2), satisfy the condition (a) in sample I or sample II. Two scenarios including selection depending on the outcome or genetic variants (Fig 3) in sample II satisfy the condition (b). The proof of this theorem is provided in S2 Text. In the case of multiple independent instrumental variables with selection (e.g., $G_i \rightarrow S$ and $G_j \rightarrow S$), the

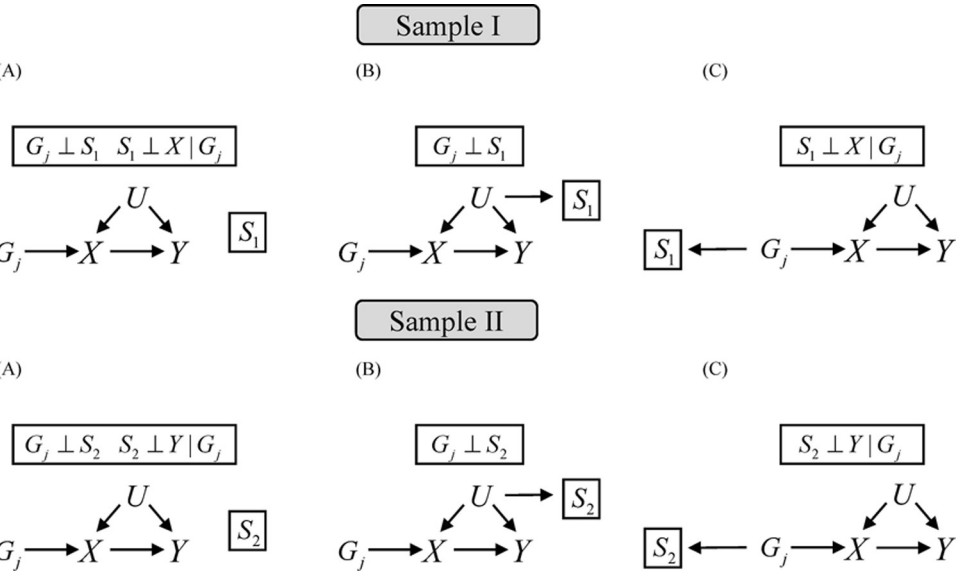

**Fig 2. Possible causal diagrams for condition (a) of Theorem 1.**

selection will result in a spurious association between $G_i$ and $G_j$. Inverse-variance weighting (IVW) with generalized least squares can reduce this bias [18].

When the exposure and outcome are binary, traditional MR methods can be used to determine whether there is a causal effect, but cannot estimate causal effect accurately [11]. In this case, the Wald ratio can be expressed as $\log(OR_{G_jY|S_2=1})/\log(OR_{G_jX|S_1=1})$. In other words, beta-

**Fig 3. Possible causal diagrams for condition (b) of Theorem 1.**

coefficients in the linear regression are replaced by log(OR)-coefficients in the logistic regression. We also provide sufficient conditions for the invariance of causal relationship using two-sample MR methods on the OR scale in Theorem 2 (S2 Text). Because the non-collapsibility [19], $S_1 \perp G_j$ and $S_2 \perp G_j$ in condition (a) are replaced by $S_1 \perp G_j | X$ and $S_2 \perp G_j | Y$, respectively. In comparison with Theorem 1, OR can avoid the influence of outcome-dependent selection bias, especially in case-control study designs [20]. The difference for the DAGs satisfying Theorem 2 is that selection depending on the unmeasured confounder no longer satisfies the condition (a) in either sample I or sample II. Instead, selection depending on exposure in sample I or outcome in sample II satisfies condition (a) in Theorem 2. Theorem 2 and its proof as well as DAGs satisfying conditions (a-b), are provided in S2 Text.

## 2.3 Two-sample MR methods

Numerous pleiotropy-robust methods have been proposed in recent years. The third core assumption of MR would be violated if pleiotropy exists, that is, the pathway between the IVs and the outcome may not be via the exposure ($X$). There are two types of pleiotropy: horizontal and vertical pleiotropy. Vertical pleiotropy is that a single nucleotide polymorphism (SNP) influences one trait, which in turn influences another. Horizontal pleiotropy occurs when SNPs influence two traits through independent pathways [21]. For example, if there are selections depending on $X$, a noncausal pathway between $G$ and $Y$ via $U$ ($G \rightarrow X \leftarrow U \rightarrow Y$) will be unlocked. This directly results in the violation of the MR assumption and further make the instrumental variables invalid. Nowadays, many premiere MR studies feature new instrument-based estimators that do not, strictly speaking, require that all proposed instruments are valid instruments. We wonder whether these methods are robust when non-random selections exist.

We consider eight methods that can be classified into four main types: consensus methods, regression-based methods, likelihood-based methods and outlier-robust methods. The consensus methods take their causal estimate as a summary measure of the distribution of the ratio estimates ($\hat{\beta}_{Y_j} / \hat{\beta}_{X_j}$), including two methods: the weighted median method [10] and the mode-based estimate (MBE) method [14]. The regression-based methods regress the genetic associations with outcome against the genetic associations with exposure using a variety of regression methods, including IVW, MR-Egger [9] and MR-robust method [12]. The likelihood-based methods include the contamination mixture method [15] and MR-Robust Adjusted Profile Score (RAPS) [16]. We also study methods that remove outliers and then estimate the causal effect of exposure on the outcome, such as MR-Lasso method [22]. The details of the eight methods are provided in S3 Text.

## 2.4 Selection mechanisms in two-sample MR

We consider seven possible selection mechanisms which depend on $G$, $X$, $Y$ and their combinations in two samples, respectively. A total of 49 nonrandom selection mechanisms are considered. The DAGs depicted in Fig 4 show the causal relationships among the variables in sample I and sample II of the MR analysis under different selection mechanisms. Fig 4A-G correspond to selection depending on $X$, $Y$, $G$, $X+Y$, $G+X$, $G+Y$, and $G+X+Y$, respectively. Note that we consider the selection mechanisms depending on $Y$ in the GWAS analysis which aims to investigate the genetic association with $X$, and vice versa. For example, nonrandom selection depends on $X$ in sample II unlocking the path $G \rightarrow X \leftarrow U \rightarrow Y$ thus misestimating the relationship of $G$-$Y$. We also consider the selection mechanism depending on $G$ because nonrandom selection is based on another phenotype that the genetic variants also affect, that is pleiotropy, but not non random selection using observed genotyping data.

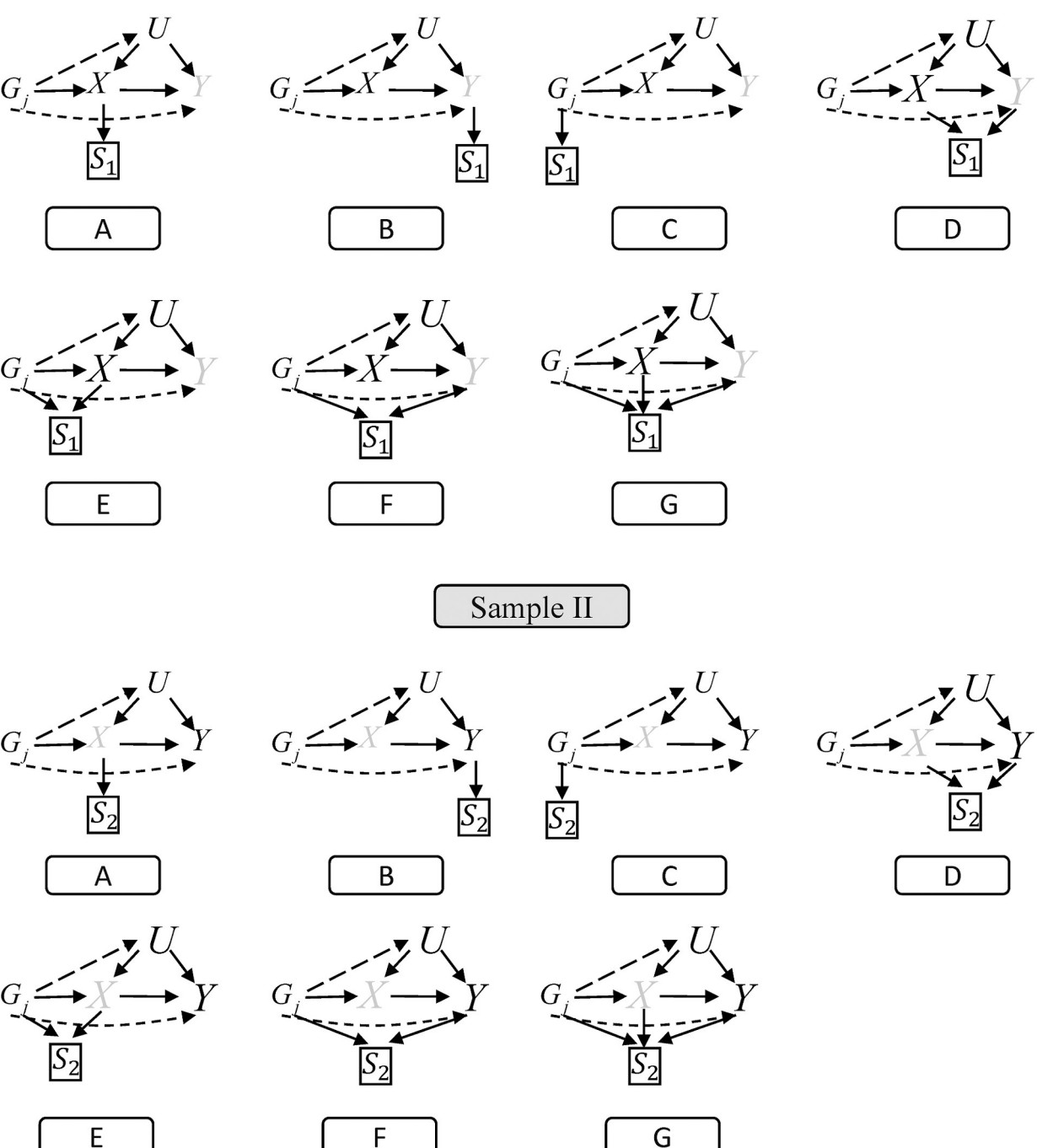

**Fig 4. Direct acyclic graphs of two-sample Mendelian randomization analysis under seven different selection mechanisms in Sample I ($S_1$) and Sample II ($S_2$), respectively.** A-G corresponding to selection depending on $X$, $Y$, $G$, $X+Y$, $G+X$, $G+X+Y$, respectively.

## 2.5 Simulation settings

In order to compare the performances of the above eight methods, we generate the following datasets as shown in Fig 4. For the $i$-th individual, we have

$$G_{i,j} \sim Binomial(2,0.3) \ j \in (1, \cdots, J),$$

$$U_i = \sum_{j=1}^{J} \phi_j G_{i,j} + \varepsilon_{U,i}$$

$$X_i = \sum_{j=1}^{J} \alpha_j G_{i,j} + bU_i + \varepsilon_{X,i}$$

$$Y_i = \sum_{j=1}^{J} \gamma_j G_{i,j} + \theta X_i + cU_i + \varepsilon_{Y,i}$$

$\varepsilon_{U,i}, \varepsilon_{X,i}$ and $\varepsilon_{Y,i} \sim N(0,1)$

$$S_i \sim Binomial(1,\pi_i) \text{ where } logit(\pi_i) = e_0 + e_x X_i + e_y Y_i + \sum_{j=1}^{J} e_{gj} G_{i,j},$$

where $e_x$, $e_y$ and $e_g$ are allowed to take values of –2, –1, –0.5, 0, 0.5, 1 and 2. The genetic variants are modelled as SNPs with a minor allele frequency of 30%, and take on values of 0, 1 or 2. The error terms $\varepsilon_{U,i}, \varepsilon_{X,i}$ and $\varepsilon_{Y,i}$ follow independent normal distributions with a mean 0 and unit variance. The selection $S$ follows a binomial distribution with selection probability depending on the exposure, outcome, and genetic variants.

We consider the following four scenarios:

1. No pleiotropy: The Instrument Strength Independent of Direct Effect assumption (InSIDE) satisfied–valid IVs with no direct effect on the outcome ($\gamma_j = 0$) and the unmeasured confounder ($\phi_j = 0$).

2. Balanced pleiotropy, InSIDE satisfied: Invalid IVs ($G_j$) with direct effects on the outcome generated from a normal distribution centered at zero, i.e. $\gamma_j \sim N(0,0.15)$, and genetic effects on the confounder are zero ($\phi_j = 0$).

3. Directional pleiotropy, InSIDE satisfied: Invalid IVs ($G_j$) with direct effects on the outcome generated from a normal distribution centered away from zero, i.e. $\gamma_j \sim N(0.1,0.15)$, and genetic effects on the confounder are zero ($\phi_j = 0$).

4. Directional pleiotropy, InSIDE violated: Invalid IVs ($G_j$) with direct effects on the outcome generated from a normal distribution centered away from zero, i.e., $\gamma_j \sim N(0.1,0.15)$, and indirect effects on the outcome via the unmeasured confounder, i.e., $\phi_j \sim U(0,0.1)$.

The causal effect of exposure on the outcome is either taken as null ($\theta = 0$) or positive ($\theta = 0.2$). Genetic associations with exposure $\alpha_j$ are drawn from a uniform distribution. Parameters are chosen such that the total proportion of variance explained in the exposure by direct effects of the genetic variants is approximately 10%. We simulate data on $J = 50$ and 100 genetic variants, and the proportion of invalid instrumental variables is 30% and 70%. We firstly generate two populations with 1,000,000 individuals, respectively. For each selection mechanism, 10,000 individuals are selected from above two populations. We generate 1,000 simulated datasets for each scenario.

In each scenario, we consider the following seven selection mechanisms (Fig 4A-G) with $e$ denoting the selection effect in sample I and sample II, respectively.

A. The selection $S$ depends on exposure ($X$), i.e. $e_x = e$, $e_y = e_{gj} = 0$;

B. The selection $S$ depends on outcome ($Y$), i.e. $e_y = e$, $e_x = e_{gj} = 0$;

C. The selection $S$ depends on genetic variants ($G$), i.e. $e_{gj} = e$, $e_x = e_y = 0$;

D. The selection $S$ depends on exposure ($X$) and outcome ($Y$), i.e. $e_{gj} = 0$, $e_x = e_y = e$;

E. The selection $S$ depends on exposure ($X$) and genetic variants ($G$), i.e. $e_x = e_{gj} = e$, $e_y = 0$;

F. The selection $S$ depends on genetic variants ($G$) and outcome ($Y$), i.e. $e_y = e_{gj} = e$, $e_x = 0$;

G. The selection $S$ depends on exposure ($X$), outcome ($Y$) and genetic variants ($G$), i.e. $e_x = e_y = e_{gj} = e$.

A total of 49 nonrandom selection mechanisms are considered. For each scenario, we assess the performances of eight pleiotropy-robust methods based on biases, SEs, type I error rates and powers. The nominal level is set to 0.05.

## 2.6 Application example

Coronary heart disease (CHD) is the leading cause of death and disability and its prevalence is increasing worldwide [23]. Its modifiable risk factors, including obesity and HbA1c play important roles in CHD prevention [24–26]. Obesity, typically defined based on body mass index (BMI), as well as waist-to-hip ratio (WHR), is a leading cause of CHD in the population. WHR adjusted for BMI (WHRadjBMI) is a surrogate measure of abdominal adiposity and has been correlated with direct imaging assessments of abdominal fat. Emdin et al. found that a genetic predisposition to higher WHR adjusted for BMI is associated with an increased risk of CHD [26]. A MR study using UK Biobank revealed that HbA1c caused CHD [25]. A Network MR analysis inferred that a higher BMI conferred an increased risk of CHD, which was partially mediated by HbA1c [24]. We aim to explore whether the causal estimation of obesity on HbA1c are different in patients with CHD and the general population. The realistic causal diagram is shown in Fig A in S4 Text. Fig A1 in S4 Text shows the DAG for sample I. Figs A2 and A3 in S4 Text are the DAGs for sample II in general population and CHD patients respectively.

We use GWAS summary data on BMI [27], WHR and WHRadjBMI [28] for European descent from the Genetic Investigation of ANthropometric Traits (GIANT) by GWAS meta-analyses of 339,224 and 224,459 individuals, respectively. GIANT is an international collaboration that seeks to identify genetic loci that modulate human body size and shape by performing meta-analysis of GWAS data and other large-scale genetic datasets. We choose the SNPs with significant association with obesity ($p<5\times10^{-8}$), minor allele frequency (MAF) more than 5% and satisfying Hardy–Weinberg equilibrium ($p>0.05$). We prune the variants by linkage disequilibrium (LD) ($r^2>0.001$).

We retrieve the individual data for HbA1c from the UK Biobank with a sample of 487,314 Europeans. The UK Biobank [29] is a prospective cohort study with rich genetic, physical and health data collected from more than 500,000 individuals (age range 40–69 years) across the United Kingdom in 2006–2010. To examine the bias of the causal effect estimation under a nonrandom selection mechanism, we also use a selected sample enriched for CHD patients with 26,765 individuals as the selected population. The HbA1c levels is measured by HPLC analysis on a Bio-Rad VARIANT II Turbo and is natural log-transformed to approximate normal distributions. CHD is defined by ICD-10 I20–I25.9 and self-reported as 1066. We performed GWAS analysis in both the general and CHD populations. Results are available for BMI, WHR and WHRadjBMI-associated leading SNPs for HbA1c. GWAS summary data for application can be found in S1 Data.

## 3 Results

### 3.1 Results of simulation

When the causal effect is zero ($\theta = 0$), Fig 5 shows the tendency of estimations under different selection mechanisms while varying across the selection effects of $X$, $Y$ or $G$ ($e_x$, $e_y$, or $e_g$) in scenario 1. Each row represents one of the seven different selection mechanisms in sample I, and the columns represent seven different selection mechanisms in sample II. The first row of Fig 5 illustrates the simulation results when the selection mechanism depends on $X$ in sample I and all selection mechanisms in sample II, respectively. The biases of all methods are negative and increase as the selection effect increases when the selection depends on $G$, $X+Y$, $G+X$, $G+Y$, $G+X+Y$ in sample II. Among these eight methods, MR-Egger shows less biases than other methods, especially when selection depends on $G+X$, $G+Y$ and $G+X+Y$. On the contrary, selections depending on $Y$ and $G$ in sample II show nearly unbiased causal effect estimations. When the selections depend on $Y$, $G$, $X+Y$, or $G+Y$ in sample I, the biases of estimations show a similar tendency as that when selection depends on $X$. However, selections depending on $G+X$ and $G+X+Y$ in sample I show different results. When the selections depend on $Y$ or $G$ in

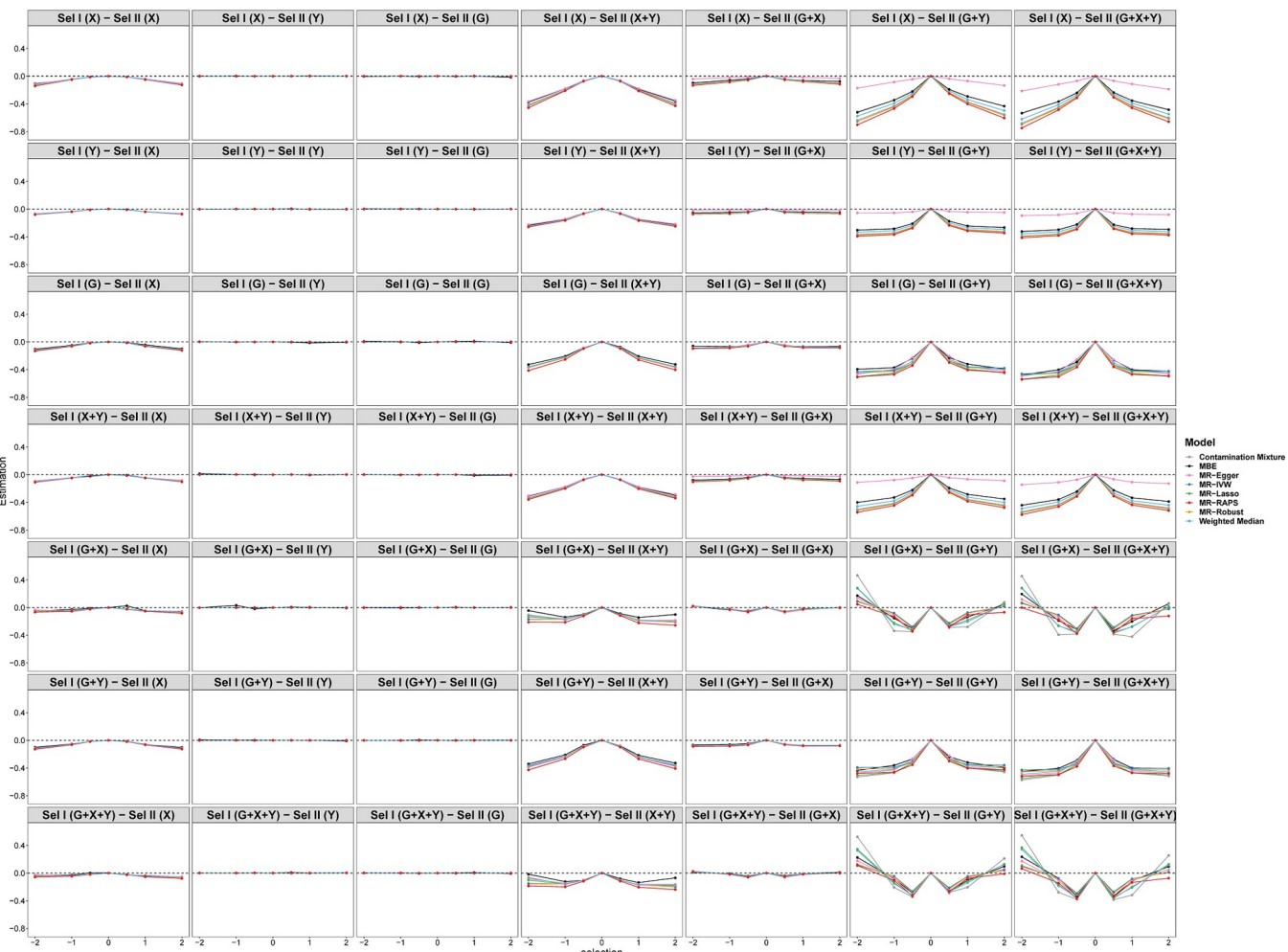

**Fig 5. Simulation results for causal estimations of eight Pleiotropy-robust MR Methods varying across selection effect from -2 to 2 under different selection mechanisms with Null causal effect in scenario 1 (50 genetic variants).** Sel I and II represent selection in sample I and sample II, respectively.

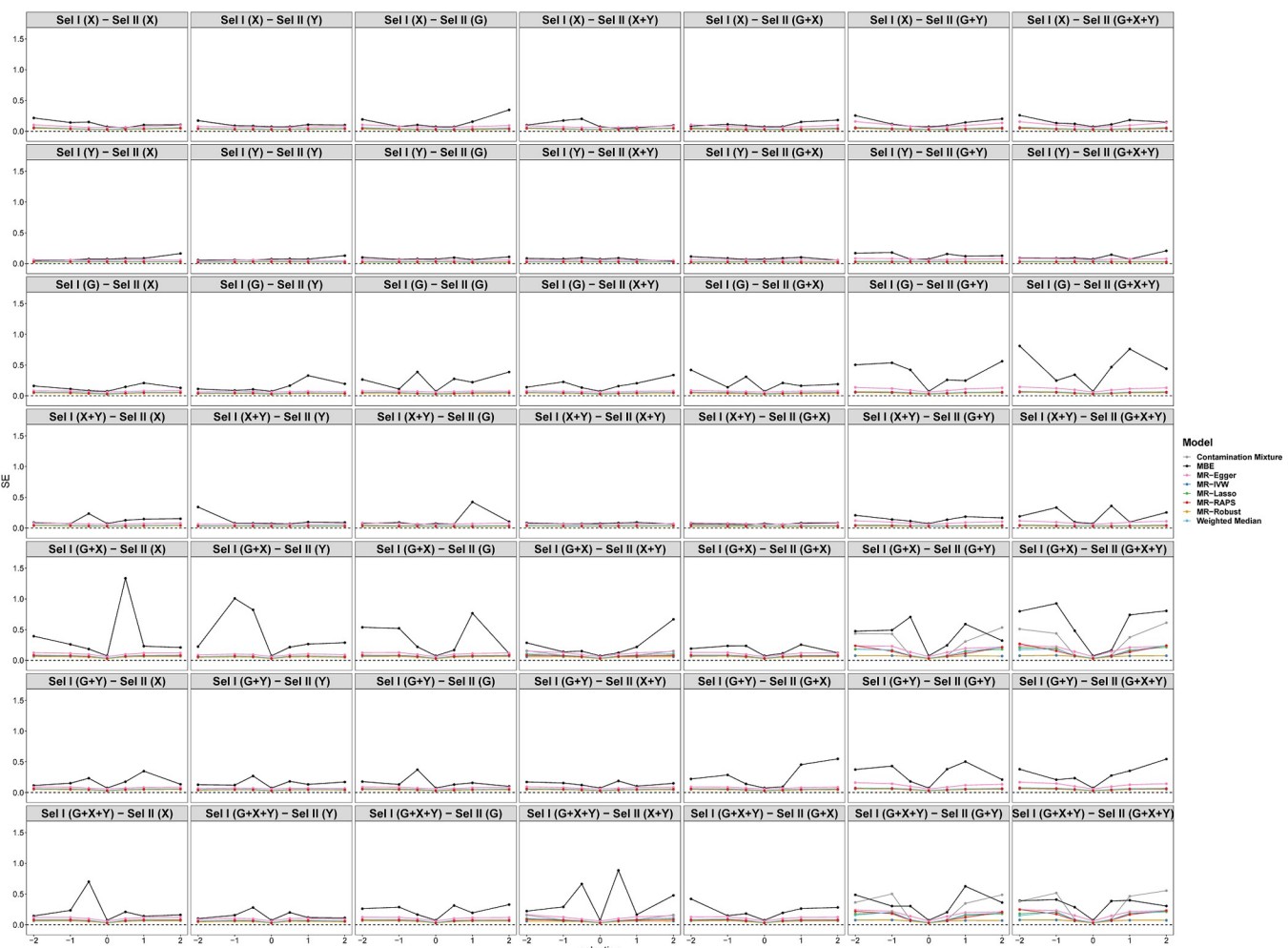

**Fig 6. Simulation results for standard error of eight Pleiotropy-robust MR Methods varying across selection effect from -2 to 2 under different selection mechanisms with Null causal effect in scenario 1 (50 genetic variants).**

sample II, all eight models also show unbiased causal effect estimations. When the selection depends on $X$, $X+Y$ and $G+X$ in sample II, the biases firstly increase then decrease with the selection effect increasing. In addition, the biases when selection depends on $G+Y$ and $G+X+Y$ firstly increase and then reduce to zero, and finally rise in the opposite direction with the selection effect increasing. In summary, the biases of selections depending on $X+Y$, $G+Y$ and $G+X+Y$ in sample II are larger than those of other selection mechanisms. When the selection mechanism in sample II is fixed, the different selection mechanisms in sample I show similar trends.

Fig 6 shows the tendency of the SEs under different selection mechanisms. In general, the SEs of the MBE model are larger than those of other models. The SEs of selection depending on $G+Y$ and $G+X+Y$ in sample II are larger than those of other selection mechanisms regardless of the selection mechanism in sample I. Fig 7 displays the tendency of type I error rates under different selection mechanisms and simulation situations. Consistent with Fig 5, the type I error rates of selections depending on $Y$ and $G$ in sample II are close to 0.05 regardless of the selection mechanism in sample I. Furthermore, the type I error inflation can be observed under other selection mechanisms due to the biased causal effect estimations of $X$ on $Y$.

When the causal effect is positive ($\theta = 0.2$), Figs 8 and 9 display a similar tendency of estimations and SEs with Figs 5 and 6. Note that when the selection depends on $Y$ in Sample I, the biases

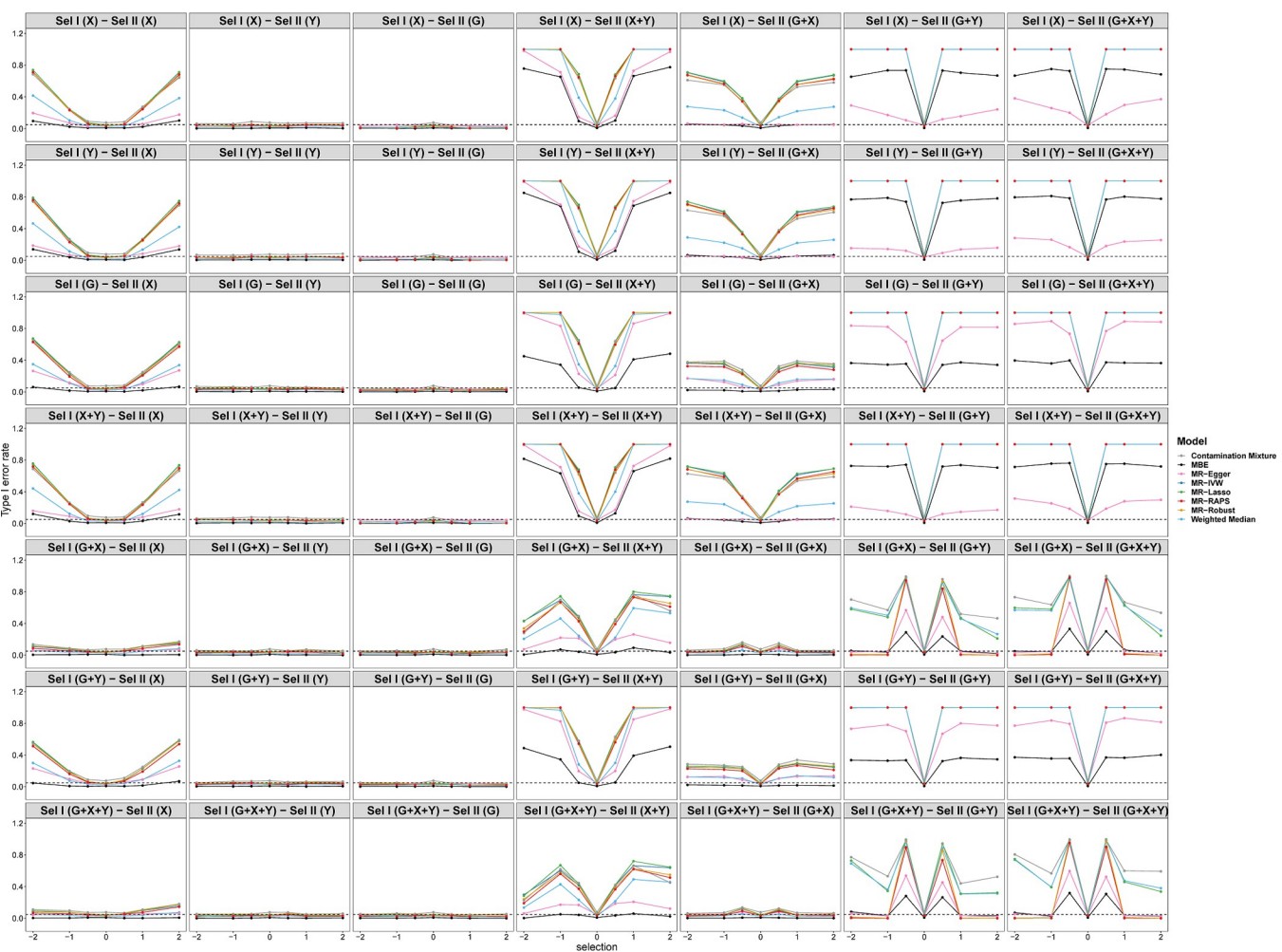

**Fig 7. Simulation results for type I error rates of eight Pleiotropy-robust MR Methods varying across selection effect from -2 to 2 under different selection mechanisms with Null causal effect in scenario 1 (50 genetic variants).**

are larger than those in the case of null causal effect. Fig 10 shows the tendency of the power under different selection mechanisms. The eight methods cannot effectively reject the null hypothesis due to nonrandom selection. For SE, MBE has the worst performance among the eight methods.

We further investigate the impact of different proportions of invalid IVs (30% and 70%), different numbers of total IVs (50 and 100 variants) and different pleiotropy scenarios (scenarios 1–4, described in section 3.1) on biases, SEs, type I error rates and statistical power. The results and details are displayed in S5–S8 Texts. When pleiotropy exists, the eight MR methods show large biases and the type I error inflation. Even in the scenarios of balanced pleiotropy, all the eight MR methods especially MR-Egger demonstrate large negative bias regardless of the null and positive causal effect. We have provided four spreadsheets (S1–S4 Tables) as supplementary materials giving the data points for all the Figs.

## 3.2 Results of application example

After the quality control process described in section 2.6, 33, 24 and 67 independent loci associated with BMI, WHR and WHRadjBMI, respectively, are included in our study. These SNPs can explain 0.41%, 0.14% and 0.15% (*F statistics* >>10) of the variance of the three exposures,

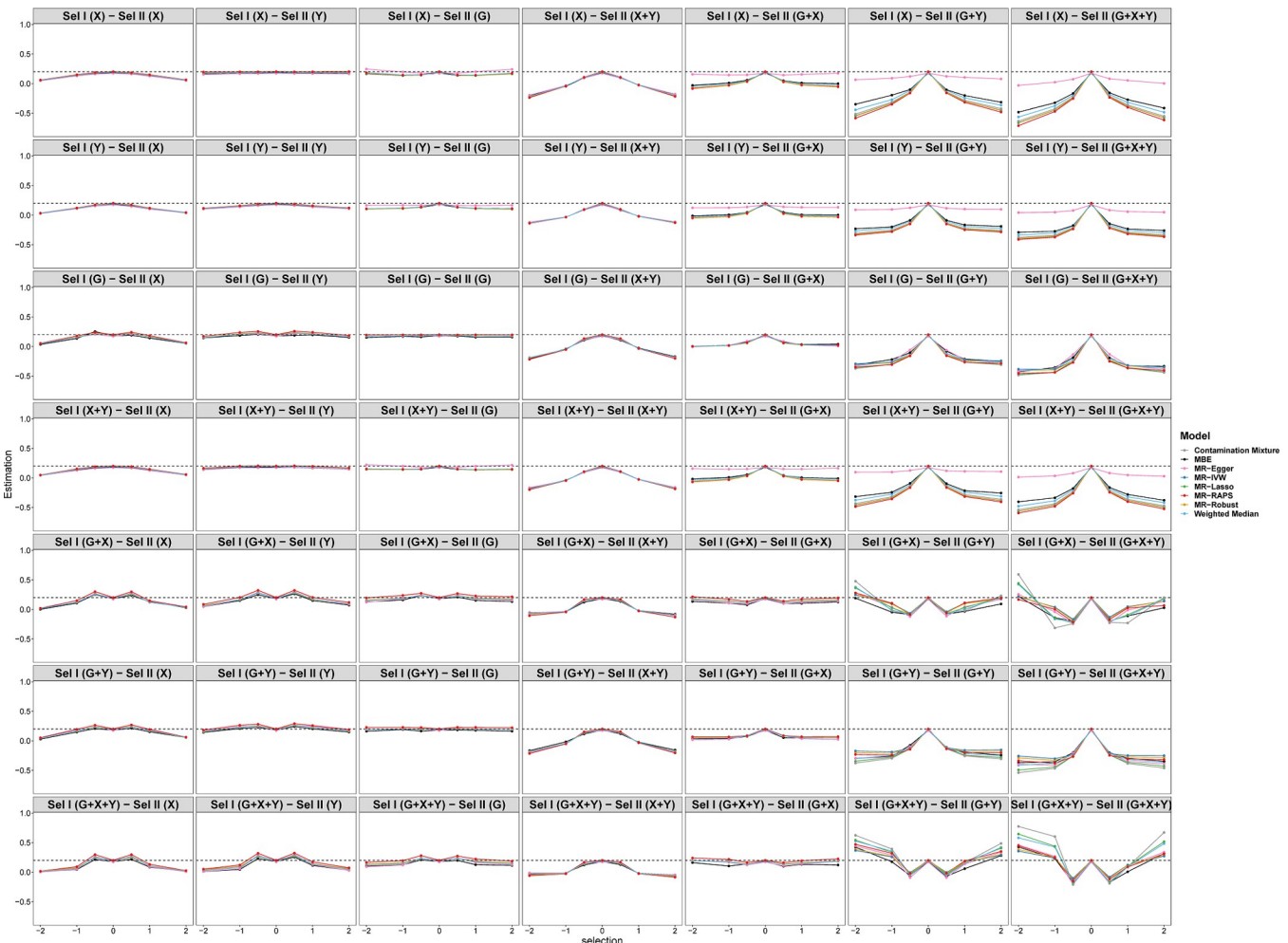

**Fig 8. Simulation results for causal estimation of eight Pleiotropy-robust MR Methods varying across selection effect from -2 to 2 under different selection mechanisms with Positive causal effect in scenario 1 (50 genetic variants).**

respectively. We use the same SNPs in the general population and the selected population, for the latter population we select individuals who are CHD patients. We then retrieve the GWAS summary results on HbA1c from the UK Biobank. We consider the causal effects of BMI, WHR, and WHRadjBMI on HbA1c levels in the general population and patients with CHD, respectively. All analyses in our study are implemented by R package TwoSampleMR.

The results are shown in Fig 11. In the general population, there are strong evidences for positive causal associations of BMI, WHR and WHRadjBMI on HbA1c levels. This means that a high BMI and WHR can improve the HbA1c levels. And the eight MR methods demonstrate consistent results. The effect estimates are magnified in the patients with CHD. This verifies that the nonrandom selection in sample II bias the effect estimation.

## 4 Discussion

The goal of this study is to explore the influence of nonrandom selection mechanisms on causal effect estimation in two-sample MR methods. Our simulation results indicate that non-random selection mechanisms will lead to substantial bias in the MR estimation and inflated type I error rates. When all the instrumental variables are valid, the different selection

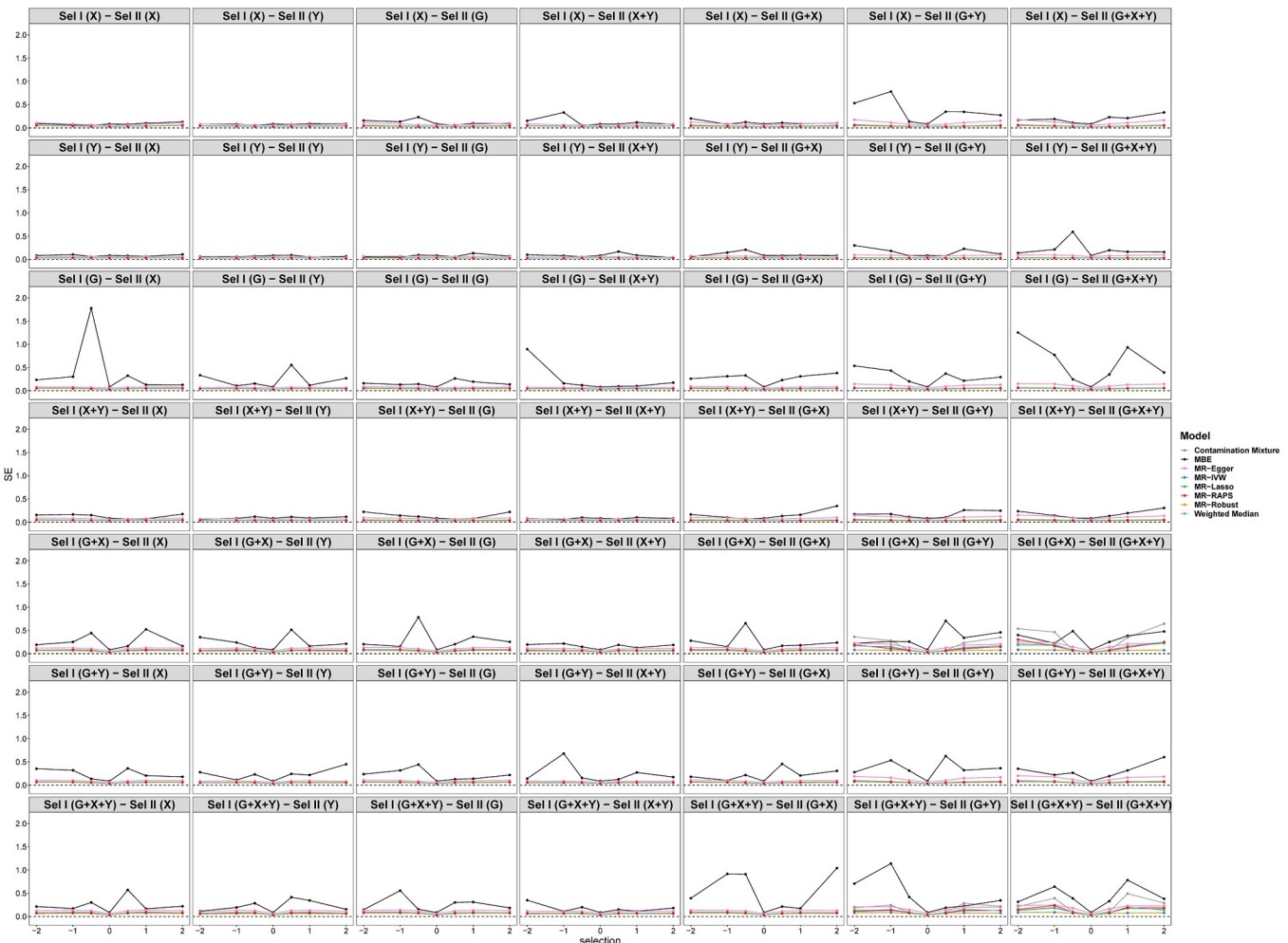

**Fig 9. Simulation results for standard error of eight Pleiotropy-robust MR Methods varying across selection effect from -2 to 2 under different selection mechanisms with Positive causal effect in scenario 1 (50 genetic variants).**

mechanism in sample II are found to have larger influence on estimation than sample I. Selections depending on the combination of $Y$ and other variables ($G$ or $X$) in sample II lead to larger biases of estimation than other selection mechanisms. The type I error inflation can be observed under 49 different selection mechanisms. Notably when the causal effect is positive, selection depending on $Y$ leads to a larger bias than the case of a null causal effect. None of the eight methods can effectively reject the null hypothesis due to selection bias. In particular, the MBE has the worst performance as its large SE. When pleiotropy exists, eight MR methods perform poorly. Even in the scenarios of balanced pleiotropy, all eight MR methods especially MR-Egger demonstrate large negative bias regardless of the null and positive causal effect.

In sample I, the nonrandom selection depending on $Y$ has less impact on the relationship between $G$ and $X$. On the contrary, the relationship between $G$ and $Y$ is largely biased the non-random selection depending on $X$ in sample II, regardless of null or positive causal effect. This is because $X$ is a collider in the pathway $G{\rightarrow}X{\leftarrow}U{\rightarrow}Y$ and $S$ is the descendant node of collider $X$. Conditioning on S also unlocks the pathway $G{\rightarrow}X{\leftarrow}U{\rightarrow}Y$ and violates the assumption of IV Independence and Exclusion restriction. To some extent, all the eight MR methods can minimize the impact of violating the assumption of Exclusion restriction. However, the IV

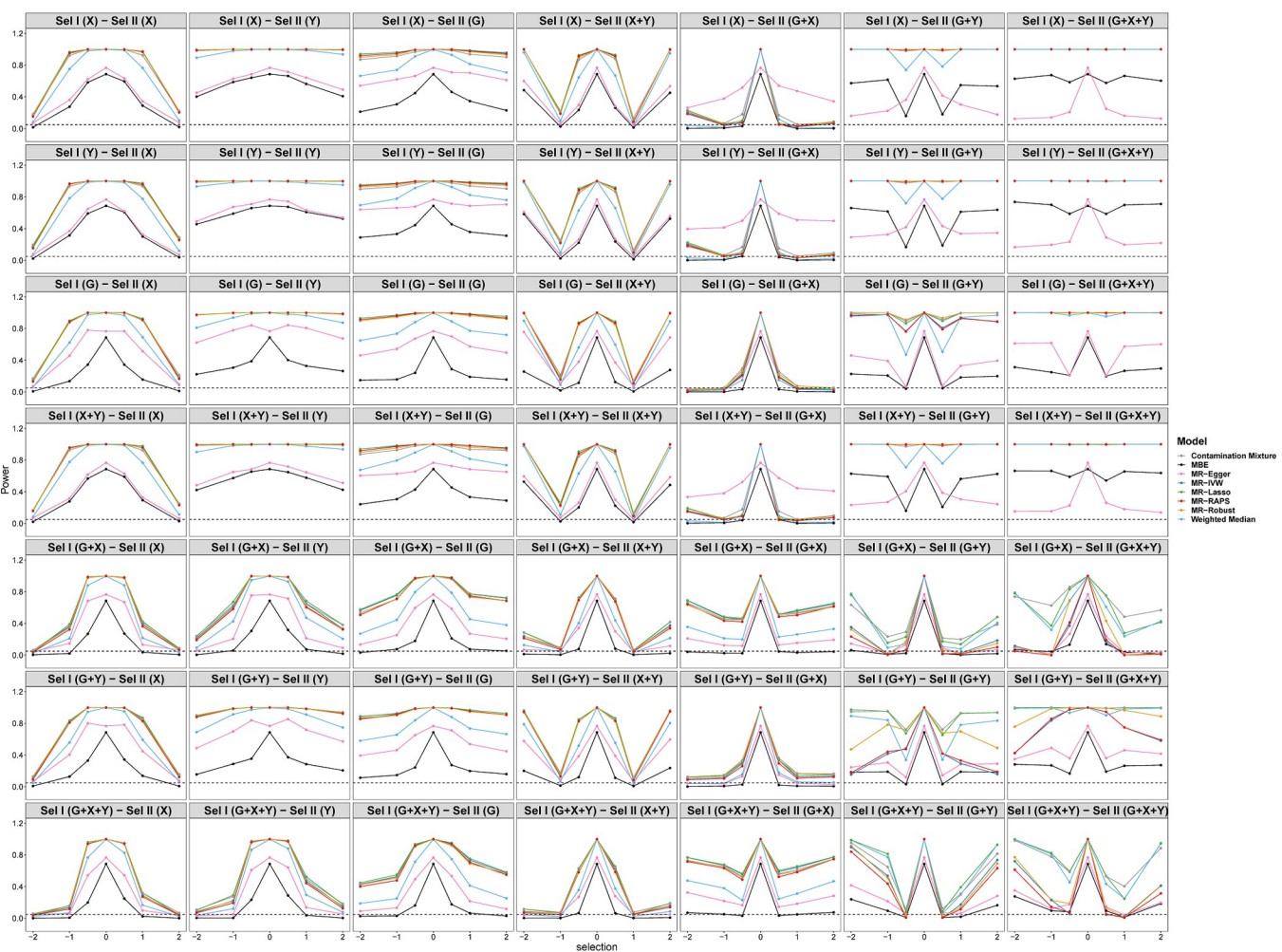

**Fig 10. Simulation results about statistic power of eight Pleiotropy-robust MR Methods varying across selection effect from -2 to 2 under different selection mechanisms with Positive causal effect in scenario 1 (50 genetic variants).**

Independence assumption is difficult to test and relax because of the unmeasured confounder *U*.

Selections depending on the combination of *Y* and other variables (*G* or *X*) in sample II lead to larger biases of estimation. Figs 5 and 8 show that the selections depending on *X+Y*, *G+Y* and *G+X+Y* in sample II lead to large biases of estimation. Selection depending on *G* and *Y* simultaneously, induces a spurious association between *G* and *Y* due to conditioning on collider *S*, that is, a horizontal pleiotropy. Selection depending on *X* and *Y* simultaneously, not only induces a spurious association between *X* and *Y*, but also unlocks the pathway *G*→*X*←*U*→*Y* [8]. Selection depending on *G*, *X* and *Y* simultaneously combines the above two cases. Hartwig et al. [30] also have found that the causal structure in the second sample had a larger influence on causal effect estimation. Their work aimed to assess whether different covariable-adjusted summary associations in two-sample MR could distort causal effect estimation. They found that using covariable-adjusted summary associations may bias the MR analyses. Particularly, the presence of an unmeasured confounder between the covariate and outcome in the second sample would render the covariate a collider. This type of collider bias is called the *analytical colliding bias* [31]. Their work is similar to our

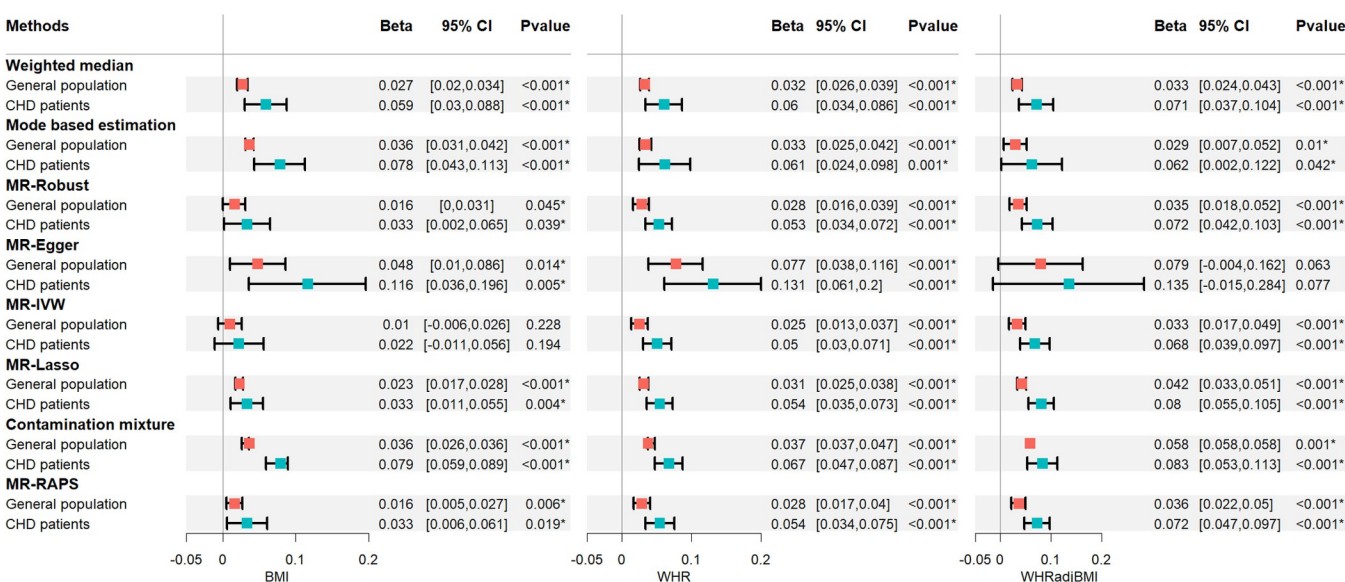

**Fig 11. Results of MR analysis of obesity on the risk of HbA1c levels.** Three columns from left to right represent MR results of body mass index (BMI), waist-to-hip ratio (WHR) and WHR adjusted for BMI (WHRadjBMI) on HbA1c levels, respectively. The red and blue nodes represent MR analysis in general population and coronary heart disease (CHD) patients, respectively.

work's conclusion in that $S$ is also a collider and restricting the analysis to the selected population will lead to another type of collider bias *sampling colliding bias* [31]. Both types of collider bias will distort the true relationship between the common causes of collider.

For all eight pleiotropy-robust methods, performance is poor under nonrandom selection mechanisms, even when the extra pleiotropy exists. When the nonrandom selection depends on multiple independent genetic variants ($G$), spurious associations among these genetic variants would be induced. This may disturb the selection of valid IVs or make the valid IVs invalid. The proportion of invalid IVs caused by nonrandom selection is difficult to measure, which influences the performance of pleiotropy-robust methods to different extents. For example, IVW requires that all the IVs are valid, weighted median allows 50% IVs are invalid, weighted MBE allows 50%-100% IVs are invalid and MR-Egger allows 100% IVs are invalid. In addition, unlocking the pathway $G \rightarrow X \leftarrow U \rightarrow Y$ violates the InSIDE assumption, which is necessary for MR-Egger [14].

We provide a short review of the relevant selection mechanisms in GWASs and examples corresponding to selection mechanisms in published two-sample MR studies, as well as existing methods to correct for selection in these situations (Table 1). Because the two samples used in MR analysis are both from GWAS analysis, we only list possible selection mechanisms in one GWAS sample, including three cases: selection depending on genetic variants, phenotype (exposure X or outcome Y in MR analysis) and both. In our application, we restrict analysis to CHD patients to reveal the significant influence of nonrandom selection on the causal effect estimation of obesity on HbA1c levels. Several MR studies have found that obesity and HbA1c levels play important roles in CHD prevention [24–26]. HbA1c is the outcome of interest and $S$ is a binary variable indicating whether a participant is a CHD patient in sample II (Fig A in S4 Text). In other words, we restrict the analysis to CHD patients, that is, conditioning $S = 1$, which depends on the outcome and exposure. In this situation, this selection mechanism can magnify the causal effect estimation of obesity on HbA1c levels.

**Table 1. Possible selections in published analysis and existing methods to correct for selection bias.**

| DAGs | Relevant selection mechanisms in GWASs | Examples corresponding to selection mechanisms in published two-sample MR studies | Selection-corrected methods |
|---|---|---|---|
| $G_j \rightarrow X/Y$<br>$\downarrow$<br>$\boxed{S}$ | Survival bias, that is, survival until study inclusion [32]; When considering a secondary disease outcome, such as disease progression [33]; Case-control study [8]; Differential loss to follow up in cohort studies [34]; Participant dropout [7]; Case-only study[7]. | Identifying risk factors for brain tumors using a case-control design [37]; A case-only study is used to explore the causal effect of BMI on breast cancer survival for ER-positive breast cancer [38]; Lp(a) was not associated with future cardiovascular mortality in the population of individuals who already established CHD [39]; Genetic associations of alcohol-related variants with esophageal cancer have been considered separately in non-drinkers, moderate drinkers and heavy drinkers [40]. | Consistent estimators of the causal relative risk and odds ratio if a priori knowledge is available regarding either the population disease prevalence or the population distribution of the IV [43]; Causal effect estimation under a quasi-empirical likelihood framework [44]; Inverse probability weighting method [7,33,45,46]; Inverse probability-of-censoring weighted estimation [34]; Bayesian approach [47,48]; Heckman's Two-Step Method [48,49]. |
| $G_j \rightarrow X/Y$<br>$\downarrow$<br>$\boxed{S}$ | A misalignment in time zero [35]; Non-random selection based on another phenotype that the genetic variants also affect. | A mendelian randomization study revealed that low blood HDL-C is a potential causal risk factor for impaired cognition during aging in non-Hispanic whites of European ancestry [41]; A Mendelian randomization study of the effects of late-life cholesterol levels on dementia risk could be biased if earlier cholesterol levels affect living long enough to be at risk of dementia [35]. | Inverse probability weighting method [7,33,45,46]; Bayesian approach [47,48]; Heckman's Two-Step Method [48,49]. |
| $G_j \rightarrow X/Y$<br>$\searrow \quad \swarrow$<br>$\boxed{S}$ | Restrict population by specific features [36]. | If a genotype and maternal folate status affect embryo failure, the selection bias will occur in the population with embryo failure. [36,42] | Inverse probability weighting method [7,33,45,46] |

In conclusion, nonrandom selection mechanisms in two-sample MR exacerbate the estimation bias for pleiotropy-robust MR methods. The biases tend to be exaggerated in the presence of pleiotropy.

## Supporting information

**S1 Text. Modeling assumptions.**
(PDF)

**S2 Text. Proof of Theorems.**
(PDF)

**S3 Text. Eight pleiotropy-robust methods in simulation.**
(PDF)

**S4 Text. DAG for application.**
(PDF)

**S5 Text. Simulation results of eight Pleiotropy-robust MR Methods with 100 valid genetic variants in scenario 1.**
(PDF)

**S6 Text. Simulation results of eight Pleiotropy-robust MR Methods in scenario 2 (Balanced pleiotropy, InSIDE satisfied).**
(PDF)

**S7 Text. Simulation results of eight Pleiotropy-robust MR Methods in scenario 3 (Directional pleiotropy, InSIDE satisfied).**
(PDF)

**S8 Text. Simulation results of eight Pleiotropy-robust MR Methods in scenario 4 (Directional pleiotropy, InSIDE violated).**
(PDF)

**S9 Text. Code to implement the method and reproduce all simulations and analyses.**
(PDF)

**S1 Table. Quantification of data shown in Figs 5–10 and A-F in S5 Text.**
(XLSX)

**S2 Table. Quantification of data shown in Figs A-X in S6 Text.**
(XLSX)

**S3 Table. Quantification of data shown in Figs A-X in S7 Text.**
(XLSX)

**S4 Table. Quantification of data shown in Figs A-X in S8 Text.**
(XLSX)

**S1 Data. GWAS summary data for application.**
(XLSX)

## Author Contributions

**Conceptualization:** Hongkai Li, Fuzhong Xue.

**Data curation:** Lei Hou, Yifan Yu.

**Formal analysis:** Yifan Yu.

**Investigation:** Xinhui Liu.

**Methodology:** Yuanyuan Yu, Hongkai Li.

**Software:** Yuanyuan Yu, Lei Hou.

**Supervision:** Xu Shi, Fuzhong Xue.

**Validation:** Lei Hou, Xiaoru Sun.

**Visualization:** Xinhui Liu.

**Writing – original draft:** Yuanyuan Yu, Lei Hou.

**Writing – review & editing:** Xu Shi, Xiaoru Sun, Zhongshang Yuan, Hongkai Li, Fuzhong Xue.

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
