## [Decision Letter · Decision Letter 0]

28 Oct 2021

Dear Dr Xue,

Thank you very much for submitting your Research Article entitled 'Impact of Nonrandom Selection Mechanisms on the Causal Effect Estimation for Two-sample Mendelian Randomization Methods' to PLOS Genetics.

We apologise for the lengthy wait for a decision, we had considerable difficulties recruiting reviewers.  There is one reviewer report attached and an editor comment that identify some concerns that we ask you address in a revised manuscript.

[LINK]

Yours sincerely,

Caroline Relton

Associate Editor

PLOS Genetics

David Balding

Section Editor: Methods

PLOS Genetics

Editor comment:

Although the standard of scientific English is generally very good, particularly in Section 3.2 there were sentences that were confusing to me, such as "... there was an evidence that it more likely to be a morning person (chronotype) were associated with a lower odds of smoking cessation. Therefore, the population enriched for smokers is a selected population depends on chronotype."  As a result, I find it hard to understand the significance of this application: are you claiming that there have been published errors?  Or just the potential for error due to selection?  This application was also not mentioned in the Discussion, making its impact unclear.  Can you point to any possible or likely errors in published analyses due to the selection effects that you describe?

Reviewer #1:

Please see attachment.

**Have all data underlying the figures and results presented in the manuscript been provided?**

Reviewer #1: **No: **Please provide a spreadsheet as supplemental material giving the data points for all the figures.

PLOS authors have the option to publish the peer review history of their article (what does this mean?). If published, this will include your full peer review and any attached files.

Reviewer #1: No

---

## [Decision Letter · Decision Letter 1]

20 Dec 2021

Dear Dr Xue,

Thank you very much for submitting your Research Article entitled 'Impact of Nonrandom Selection Mechanisms on the Causal Effect Estimation for Two-sample Mendelian Randomization Methods' to PLOS Genetics.

The manuscript was evaluated at the editorial level and by the independent peer reviewer. Although we appreciate the improvements in the revised manuscript, we felt that insufficient care and attention to detail was made in preparing the revision.  The reviewer has identified a number of issues that remain to be addressed.  We ask you to address these in a further revised manuscript.  The required changes should not take very much effort, but we ask you to take care to check all aspects of the manuscript for clarity and accuracy, to create what we hope will be essentially a final version.

Please also:

[LINK]

Yours sincerely,

Caroline Relton, PhD

Associate Editor

PLOS Genetics

David Balding

Section Editor: Methods

PLOS Genetics

Reviewer's Responses to Questions

Reviewer #1: The authors have responded to my comments and improved the paper. I have some further comments arising.

1. Table 1. You mention collider-corrected regression as a method to correct for selection bias, but it is not.

2. Table 1. it is not clear how population stratification and dynastic effects lead to selection bias. They are well known as confounders of genetic associations, but selection effects are not so clear; please explain.

3. Table 1 row 1 col 3. ref 48 does not use inverse probability weighting.

4. The author summary could be improved, it is not really at a lay level, X Y and G are undefined, etc.

5. Page 5 line 92. "(pleiotropy) always leads to the violation..." is incorrect, it can do, but not always.

6. Equation 1 and Figure 1. Why are gamma and phi both shown by grey dotted lines in the figure, but only gamma appears in the equation?

7. Page 8 line 157. "Existence of causal relationship" looks wrong, I think you mean something like invariance of causal relationship.

8. Page 8 line 174. IVW is undefined at this point. Not clear what bias you are referring to.

9. Page 11 line 232. "Ratio" should be "proportion".

10. Page 12 lines 250-254. The discussion of circadian rhythm relating to chemotherapy is distracting from the main question of chronotype as a cause of breast cancer. Not clear that this is necessary.

11. Page 15 line 326. "To avoid the influence of winner's curse" is poor wording - it is still possible among these selected SNPs. The point in the earlier review was that winner's curse effects would be different between two analyses using different numbers of SNPs. This clause could be deleted without detriment.

12. The smoking example is unconvincing. You claim that smoking is a descendant of the collider at chronotype. But there is no evidence that chronotype causes smoking. It may be smoking that causes chronotype, or their association could be due to confounding. Furthermore there may simply be an effect modification such that the chronotype-breast cancer association is different for smokers. The attenuated associations in the smokers are consistent with selection bias, but are not evidence for it. Finally the association reported by Gibson et al was with smoking cessation, not smoking itself.

**Have all data underlying the figures and results presented in the manuscript been provided?**

Reviewer #1: Yes

PLOS authors have the option to publish the peer review history of their article (what does this mean?). If published, this will include your full peer review and any attached files.

Reviewer #1: No

---

## [Editor Report · Decision Letter 2]

16 Feb 2022

Dear Dr Xue,

We are pleased to inform you that your manuscript entitled "Impact of Nonrandom Selection Mechanisms on the Causal Effect Estimation for Two-sample Mendelian Randomization Methods" has been editorially accepted for publication in PLOS Genetics. Congratulations!

Yours sincerely,

Caroline Relton, PhD

Associate Editor

PLOS Genetics

David Balding

Section Editor: Methods

PLOS Genetics

**Data Deposition**

http://datadryad.org/submit?journalID=pgenetics&manu=PGENETICS-D-21-00846R2

**Press Queries**

---

## [Editor Report · Acceptance letter]

4 Mar 2022

PGENETICS-D-21-00846R2 

Impact of Nonrandom Selection Mechanisms on the Causal Effect Estimation for Two-sample Mendelian Randomization Methods 

Dear Dr Xue, 

We are pleased to inform you that your manuscript entitled "Impact of Nonrandom Selection Mechanisms on the Causal Effect Estimation for Two-sample Mendelian Randomization Methods" has been formally accepted for publication in PLOS Genetics! Your manuscript is now with our production department and you will be notified of the publication date in due course.

With kind regards,

Livia Horvath

PLOS Genetics

On behalf of:
